# The Impact of Competitive Sports on Oral Health: Exploring Their Relationship with Salivary Oxidative Stress in Children

**DOI:** 10.3390/healthcare11222927

**Published:** 2023-11-08

**Authors:** Mădălina Nicoleta Matei, Paul Șerban Popa, Antonela Magdalena Covaci, Oana Chipirliu, Kamel Earar, George Stoica, Andreea Eliza Zaharia, Nicoleta Maricica Maftei, Gabriela Gurău, Elena Lăcrămioara Lisă, Anamaria Zaharescu

**Affiliations:** Research Centre in the Medical-Pharmaceutical Field, Faculty of Medicine and Pharmacy, Dunărea de Jos University of Galați, 800181 Galați, Romania

**Keywords:** glutathione peroxidase (GPX), total antioxidant capacity (TAC), superoxide dismutase (SOD), oral hygiene, oral health, periodontal disease, dental carries, competitive sports

## Abstract

This article explores the correlation between salivary biomarkers, such as glutathione peroxidase (GPX), total antioxidant capacity (TAC), and superoxide dismutase (SOD), and their association with oral health for children in competitive sports. Saliva has emerged as a valuable resource for evaluating physiological and pathological conditions due to its non-invasive collection method and easy storage. This study examines the potential of GPX, TAC, and SOD as salivary biomarkers for assessing the impact of competitive sports on children’s oral health. It discusses the potential implications of increased oxidative stress due to intense physical activity and the role of antioxidant defense mechanisms in maintaining oral health. In total, 173 children aged between 6 and 17 were divided into three groups, 58 hockey players, 55 football players, and 60 in the control group, and examined to assess their oral hygiene and dental and periodontal health. Saliva was collected, centrifuged, and the supernatant was analyzed for the relevant biomarkers. The findings seem to suggest that salivary biomarkers, like GPX, TAC, and SOD, might serve as indicators of the physiological response to competitive sports in children, as well as indicators of oral health, especially dental cavities, and periodontal disease. Statistical analysis showed significant differences between the groups, with better values for athletes, regardless of age, sex, or activity type. Understanding the relationship between salivary biomarkers and competitive sports in children can have significant implications for monitoring and optimizing the health and performance of young athletes. Further research is needed to establish the specific associations between these biomarkers and the effects of several types and intensities of sports activities on oral health in children.

## 1. Introduction

Competitive sports play a significant role in the overall health and well-being of children. However, the impact of competitive sports on oral health and salivary oxidative stress remains relatively unexplored. This article aims to bridge this gap by investigating the correlations between competitive sports, oral health, and salivary oxidative stress in children, focusing on oral hygiene, periodontal disease, and dental carries.

Sports can have both direct and indirect effects on the oxidative stress levels in athletes. Several studies have investigated the relationship between sports and oxidative stress markers, providing valuable insights into this topic. One study examined the responses to oxidative stress induced by chronic and acute exercise in rats [1]. The researchers investigated the biomarkers of oxidative stress and endogenous antioxidants in various tissues, including the brain, liver, heart, kidney, and muscles. They found that both chronic and acute exercise led to oxidative stress in these tissues, indicating that exercise can directly impact oxidative stress levels. Another study focused on elite basketball players and examined variations in oxidative stress markers at the beginning and end of a season [2]. The researchers found that intense competition during the athletic season induced oxidative stress in these athletes. This suggests that the demands of competitive sports can indirectly affect oxidative stress levels in athletes. A study investigated the oxidative stress indices in child swimmers and found increased oxidative stress markers in their blood [3]. Gender-specific oxidative stress parameters were examined in a study by Dopsaj in 2010 [4]. The study found a positive relationship between training experience and oxidative stress markers in both male and female athletes. This suggests that the training experience can influence oxidative stress levels in athletes, regardless of gender. Regular physical training seems to reduce the oxidative stress associated with exercise, as trained athletes show less evidence of oxidative stress for a given bout of exercise [5]. This suggests that regular training can enhance the defense system of athletes against oxidative stress.

Oral oxidative stress refers to an imbalance between the production of reactive oxygen species (ROS) and the body’s antioxidant defense mechanisms in the oral cavity. It is a concept that has gained significant attention in the field of redox biology and medicine [6]. Oxidative stress can have detrimental effects on oral health and has been implicated in various oral diseases, including periodontal diseases, oral mucosal diseases, and oral cancer [7,8,9,10,11,12]. Antioxidant defense mechanisms play a crucial role in maintaining redox homeostasis in the oral cavity. Antioxidant enzymes, such as glutathione peroxidase and superoxide dismutase, are responsible for scavenging ROS and protecting oral tissues from oxidative damage [6]. However, in certain oral diseases, the activity of antioxidant defense systems may be compromised, leading to increased oxidative stress. For example, patients with periodontal disease have been found to have lower levels of antioxidants and higher levels of oxidative stress biomarkers compared to healthy individuals, regardless of antibiotic therapy [13,14].

Several studies have investigated the salivary biomarkers of oxidative stress in different populations, including children. One study examined the antioxidant barrier, redox status, and oxidative damage in saliva and blood samples of healthy individuals at different ages, including children. The study found that little is known about the salivary redox balance in healthy children, adults, and the elderly. The authors highlighted the importance of understanding salivary oxidative stress in different age groups [15]. Another study focused on the salivary markers of oxidative stress in relation to periodontal and dental status in children. The authors emphasized the non-invasive nature of saliva sampling, which makes it particularly useful in research on children. They also noted that most studies on salivary markers of oxidative stress have been conducted in adult patients, and it is currently unknown whether the findings can be extrapolated to children [16]. In another study, the authors assessed oxidative stress in the saliva of children with dental erosion. They found no correlation between total antioxidant status (TAS) and oxidative stress in the saliva of children with severe dental erosion. However, the study had limitations due to a small sample size [17]. In addition, a different study highlighted the association between salivary redox balance disturbances and oxidative stress-related oral complications, such as dental caries, gingivitis, periodontitis, oral mucosa ulceration, candidiasis, and burning mouth syndrome [18]. The study emphasized the importance of maintaining the redox balance in saliva for oral health and the increased risk of oral maladies in individuals with disturbances in this balance. These studies provide evidence for a link between salivary oxidative stress and oral health. Oxidative stress can lead to DNA damage, cell injury, and morphological changes in the salivary glands, which can affect the quality and quantity of saliva produced. Disturbances in the redox balance in saliva can increase the risk of oxidative stress-related oral maladies. Therefore, maintaining a healthy redox balance in saliva is crucial for oral health. Overall, research on salivary oxidative stress in children is still limited compared to studies in adults. However, the available evidence suggests that salivary biomarkers of oxidative stress may have diagnostic potential for both oral and systemic diseases.

GPX is an enzyme that catalyzes the reduction of hydrogen peroxide and lipid peroxides, thereby protecting cells from oxidative damage [19]. In periodontal disease, the presence of bacteria, such as *Porphyromonas gingivalis*, triggers an immune response that leads to the release of cytokines and the activation of polymorphonucleocytes (PMNs). These PMNs produce ROS as part of the host’s response to infection. The increased number and activity of PMNs in patients with periodontal disease result in the release of a high amount of ROS, leading to oxidative damage to the gingival tissue, periodontal ligament, and alveolar bone [16]. GPX helps to counteract this oxidative damage by converting hydrogen peroxide into water, thus reducing the harmful effects of ROS [20]. In the case of dental caries, oxidative stress has also been implicated in its etiology and pathogenesis. The increased production of ROS in dental caries can lead to oxidative damage to tooth structures [21]. SOD has been studied in relation to oral health, particularly in the context of oral squamous cell carcinoma (OSCC), a type of oral cancer [22]. Additionally, SOD has been studied in relation to periodontal disease. Studies have shown that SOD levels can be altered in periodontal disease, indicating its potential as a biomarker for this condition [23,24]. SOD is one of the key antioxidants present in saliva, and its levels have been found to be lower in patients with periodontitis compared to healthy individuals. This decrease in SOD activity may contribute to the imbalance between oxidants and antioxidants in the oral cavity, leading to increased oxidative damage and tissue destruction in periodontal disease [25,26]. In addition to periodontal disease, SOD has also been implicated in dental caries. Studies have shown that SOD activity is reduced in patients with dental caries compared to healthy individuals. This decrease in SOD activity may impair the antioxidant defense system in the oral cavity, making the teeth more susceptible to acid-induced demineralization and the development of dental caries [27,28]. The total antioxidant capacity (TAC) measures the free radical-neutralization ability of the endogenous antioxidant system, comprised of a sulfhydryl group (primarily albumin), carotenoids, ascorbate, bilirubin, α-tocopherol, retinol, urate, and some proteins. Studies have shown that TAC levels can be altered in periodontal disease, indicating its potential as a biomarker for this condition [29,30]. In the context of dental caries, TAC has been found to be higher in individuals with active caries compared to caries-free subjects. This suggests that TAC may influence the development and activity of dental caries and can be measured using salivary factors. Regarding periodontal disease, studies have shown that salivary TAC is lower in periodontitis patients compared to healthy controls. There is also a significant negative correlation between salivary TAC and clinical attachment loss in periodontitis patients. Plasma and salivary TAC levels have been found to be inversely related to the severity of chronic periodontitis, and scaling and root planning can restore TAC levels to normal [31].

## 2. Materials and Methods

The primary objective of this research study is to investigate whether there are any disparities in salivary oxidative stress levels between children who participate in competitive sports and those who do not engage in such activities. To achieve this, three groups were established and compared: one group consisting of football players, one group of ice hockey players, and a control group, made up of children not engaging in competitive sporting activities, with the aim of identifying potential differences. The null hypothesis (H0) posits that there is no variability between the two groups. The focus of this study was to compare the salivary levels of GPX (glutathione peroxidase), SOD (superoxide dismutase), and TAC (total antioxidant capacity) in football and hockey players, in comparison to a control group with similar demographic characteristics. In choosing the sports, the key factors were the popularity among children and the similarity of physical effort type and length of training sessions or matches. Football and ice hockey best suited these criteria.

This cross-sectional study, conducted using the STROBE guidelines, involved 173 children, aged between 6 and 17. There were 58 hockey players, 55 football players, and 60 children in the control group. Based on their age and sporting activity, the children were divided into distinct groups for more relevant statical analysis. The following groups were the result: H (all hockey players), F (all football players), C (control group), H1 (29 hockey players aged 6–12), H2 (29 hockey players aged 13–17), F1 (27 football players aged 6–12), F2 (28 football players aged 13–17), C1 (30 control individuals aged 6–12) and C2 (30 control individuals aged 13–17). The reason for this particular age separation is that the physiological age of the onset of permanent dental eruption is 6 years old, while its completion is 12 years old.

Any child who received treatment with antibiotics, vitamins or dietary supplements, or professional products containing fluoride (both gel and varnishes) during the month prior to the examination was excluded from the study, regardless of their potential category. The existence of ulcers, herpes, canker sores, or tumor lesions, was also an exclusion criterion. Any chronic condition requiring ongoing treatment also rendered the individual ineligible to participate. Children undergoing orthodontic treatment were allowed to participate in the study primarily because of the lack of evidence suggesting any significant influence that this type of treatment could have on salivary oxidative stress, as highlighted by numerous articles [32,33,34]. In this study, only 7 participants out of 173 were undergoing orthodontic treatment at the time, insufficient to potentially alter the statistical analysis. All of them had fixed appliances (brackets). The children taking part in competitive sports had to be between 6 and 17 years old and to be members of an accredited sports organization for at least one year prior to the examination. For the young athletes, the presence of only deciduous teeth was an exclusion criterion. Subjects that met the eligibility criteria were recruited from two community sports clubs. Membership to these sports clubs involves at least two mandatory training sessions per week. Failing to attend the minimum number of practice sessions, without medical reasons, results in an automatic loss of membership status. Therefore, all athlete children participating in this study partook in at least two weekly sessions. Members of the control group were selected from the outpatient clinics of the local emergency clinical hospital for children and of the Faculty of Medicine and Pharmacy of the University of Galați. For both categories of children, athletes or not, participation in this study was voluntary, any interested child could have been enrolled by their parents or legal guardians, as mentioned in the consent form, which was later clinically evaluated, recording the relevant data for assessing whether the eligibility criteria had been met or not.

All the necessary examinations, including clinical assessments, saliva collection, and the determination of oral health indices, were conducted in the aforementioned facilities. To ensure consistency and accuracy, only two trained professionals performed these procedures, Prof. Dr. Mădălina Nicoleta Matei and Dr. Paul Șerban Popa. This approach was necessary due to the subjective nature of assessing oral health indices. Demographic data was obtained from the legal guardians but was not used as a comparison criterion in this study, due to its limited number of participants not meeting the statistically relevant requirements. To identify and quantify pathological clinical aspects related to the oral cavity, several of the most frequently used indices were calculated: the OHI-S (Oral Hygiene Index-Simplified) index, PMA (Papilla-Margin-Attached) gingival index, and DMF-T (Decayed-Missing-Filled Teeth) index. These indices were used to assess various aspects of oral health and provide a comprehensive understanding of the participants’ oral health status. The DMFT index is widely used worldwide. It provides an estimation of the dental disease in a population by counting the number of decayed, missing, and filled teeth. However, it has limitations, such as its failure to consider non-cavitated lesions. The ICDAS index, on the other hand, is a more comprehensive system that evaluates both cavitated and non-cavitated carious lesions. It provides a more detailed assessment of caries’ status and severity.

The participants were usually examined around noon, after finishing their school activities, they were instructed not to brush their teeth after that day’s morning routine until the examination, and were prohibited from using mouthwash and dental floss for their morning routine, as well as from eating or drinking anything except still water for at least one hour prior to sampling. Rinsing one’s mouth with tap water for 30 s prior to sampling was mandatory. The collection method involved allowing unstimulated saliva to accumulate on the floor of the mouth from where it was collected using transfer pipettes, every 60 s until 5 mL was obtained [35,36,37]. The samples were homogenized using a vortex mixer and then divided into 1 mL batches in individual Eppendorf tubes. Each tube was then centrifuged at 1500× *g* for 2 min and the supernatant collected and stored individually at −80 °C, to preserve the biomarkers and prevent degradation [38,39].

The biomarkers used to assess the salivary oxidative stress were glutathione peroxidase (GPX), superoxide dismutase (SOD), and the total antioxidant capacity (TAC). Colorimetric methods were utilized, using a TECAN infinite F50 microplate reader with Magellan Tracker v. 7.2 analysis software. GPX activity was investigated by directly measuring NADPH consumption in the enzyme-coupled reactions using a Glutathione Peroxidase Assay Kit from Sigma-Aldrich, St. Louis, MO, USA (MAK437). SOD levels were determined using Water Soluble Tetrazolium Salts-1 that produce a water-soluble formazan dye upon reduction with superoxide anion. The rate of the reduction with a superoxide anion is linearly related to the xanthine oxidase (XO) activity and is inhibited by SOD, for which a SOD Activity Kit (MAK379) from Sigma-Aldrich was used. For the TAC, a Total Antioxidant Capacity Assay Kit from abcam (ab65329) allowed measurements of both small molecule antioxidants and proteins, by converting the Cu^2+^ ion to Cu^+^, and further chelating it with a colorimetric probe proportional to the total antioxidant capacity.

Statistical analysis of the data was performed using MedCalc software version 20.215. The normal distribution was assessed using the Chi-squared test. The differences were statistically analyzed with the paired samples *t*-test. The minimum sample size was calculated using the following criteria: Type I error/Alpha (Significance) = 0.2 and Type II error/Beta (1 − Power) = 0.1. The inter- and intra-examiner reproducibility was not calculated due to the relatively small number of participants and due to the fact that only two adequately trained professionals performed all the examinations, including clinical assessment, saliva collection, and oral health indices determination. Index values’ calculation have rather clear assessment criteria and the biomarkers were determined automatically. Should a larger population need to be examined or a higher number of examiners required, the Fleiss kappa (adaptation of Cohen’s kappa for 3 or more raters) would have been calculated.

## 3. Results

For each participant, oral health indices (OHI-S, PMA, and DMF-T) were clinically determined and from the individual collected saliva sample, three oxidative stress biomarkers were determined (GPX, SOD, and TAC). Given the study’s design and criteria, nine pairs of data series were generated and statistically analyzed. The series were subjected to Chi-squared test to analyze the normality of the distribution, and paired sampled *t*-tests were conducted to establish the statistical significance of the data obtained. The relevant gathered demographic data regarding the participants are presented in the table below (Table 1).

Results and correlations between the subgroups (gender, age groups, etc.) were not presented due to the limited number of participants; statistical analysis would not have met the minimum number of participants required to determine relevant differences.

### 3.1. Oral Health Indices

Initially developed by Green and Vermillion in the 1960s and later modified in 1964, the OHI-S index was established to ascertain the key etiological factors of oral health degradation: plaque and calculus. The most recent version is provided by the World Health Organization (WHO) [40]. The OHI-S index ranges from 0 to 6, with lower values indicating better oral hygiene. The PMA gingival index uses inflammation as a periodontal health indicator [41]. Periodontal diseases first affect the papilla (P), and then progress towards the gingival margin (M) and onwards to the attached tissues (A). The inflammatory response is directly related to the severity of the disease. As the index value increases, it indicates a higher severity of periodontal pathology. The DMF-T index is used to describe the prevalence of dental caries in a population group. Like the other indices, a lower value indicates less carious activity [42]. The assessment criteria are well established, making the DMF-T index one of the most utilized carries-related indices. All three indices (OHI-S, PMA gingival index, and DMF-T index) demonstrated lower (better) values in the sport-performing groups compared to the control group. No significant differences were found among different age groups. A slight bias towards hockey players compared to children playing football was observed, but due to the limited sample size, the probability of this bias is not statistically significant. Oral health indices’ values for each group are presented below, in Table 2, while the statistical analysis and comparison is presented in Table 3.

### 3.2. Salivary Oxidative Stress Biomarkers

The glutathione peroxidase (GPX) levels for group H ranged from 435.351 to 691.694. Group F had a minimum value of 355.943 and a maximum value of 588.25. The control group registered the lowest value of 149.129, with their highest being equal to 284.185. All children involved in competitive sports had statistically significantly higher levels of GPX, regardless of age group, with *p* < 0.0001 for every category. Comparing the study participants involved in hockey to those playing football, the former had significantly higher values in total, as well as in the 13–17-year-old range; the 6–12 age range, in comparison, had a *p* = 0.0011, a less significant value.

Group’s H superoxide dismutase (SOD) activity levels range from 7.904 to 49.943. Group F had a lowest value of 0.071, and a highest value of 81.639. The control group obtained a minimum value of 34.596, and a maximum value of 90.594. All athletes showed significantly lower SOD values compared to the control group, age notwithstanding. No significant differences were obtained between hockey players and footballers, regardless of age group.

The total antioxidant capacity (TAC) assessment for hockey players revealed a minimum value of 0.125, and a maximum of 0.949. The children involved in competitive football had a lowest value equal to 0.26, with the highest being 1.496. Data for the control group ranged from 0.078 to 2.506. Both groups with children involved in competitive sports showed significant differences when compared to the control group. Taking age into consideration, the analysis revealed better significance for the 13–17-year-olds as opposed to the 6–12 age interval. Due to the sample size, higher probabilities were obtained while comparing hockey players to football players (Table 4). 

## 4. Discussion

Competitive sports have been shown to have a significant impact on the overall health and well-being of children [43]. However, the relationship between competitive sports and oral health, specifically salivary oxidative stress, has not been extensively studied. This article aimed to fill this gap by investigating the correlations between competitive sports, oral health, and salivary oxidative stress in children, with a focus on oral hygiene, periodontal disease, and dental caries.

Several studies have investigated the salivary biomarkers of oxidative stress in different populations, including children. One study highlighted the importance of understanding salivary oxidative stress in different age groups, including children [38]. Another study focused on the salivary markers of oxidative stress in relation to periodontal and dental status in children, emphasizing the non-invasive nature of saliva sampling and the need for more research in this area [44]. Another study assessed oxidative stress in the saliva of children with dental erosion and found a correlation between the total antioxidant status (TAS) and oxidative stress [45]. Additionally, a study highlighted the association between salivary redox balance disturbances and oxidative stress-related oral complications, such as dental caries and periodontitis [17]. These studies provide evidence for a link between salivary oxidative stress and oral health.

In this study, the researchers aimed to determine whether there were any differences in salivary oxidative stress and oral health between children involved in competitive sports and those not engaged in such activities. The study included 173 children aged between 6 and 17, divided into two groups of hockey players and football players, as well as an additional control group. The researchers assessed oral health indices, including the Oral Hygiene Index-Simplified (OHI-S), PMA gingival index, and DMF-T index, as well as the salivary levels of GPX, SOD, and TAC.

The results showed a significant positive influence of regular physical activity on children’s salivary oxidative stress, in accordance with other studies [15,46,47]. The results of the study showed that children involved in competitive sports had significantly higher levels of GPX compared to the control group, regardless of their age group. This suggests that regular physical activity in competitive sports may have a positive influence on children’s salivary oxidative stress. Additionally, TAC levels were significantly higher in children involved in competitive sports compared to the control group, suggesting a greater overall antioxidant capacity in these individuals. Furthermore, the low levels of SOD activity that were discovered are in line with previous studies, linking periodontal disease and dental caries with an imbalance between the production of reactive oxygen species and the body’s antioxidant defense mechanisms [45,48,49].

This study also assessed oral health indices, including the Oral Hygiene Index-Simplified (OHI-S), PMA gingival index, and DMF-T index. Regarding dental carries, the choice was between the DMF-T and ICDAS indices. Both the DMFT and ICDAS indices are commonly used for assessing dental caries. The DMFT index provides a simple estimation of dental disease in a population, while the ICDAS index offers a more comprehensive assessment of caries’ status and severity. Given our intention to assess the impact of competitive sports on children’s oral health as a whole, not focusing only on dental carries, the DMFT index was preferred, as it provides a sufficient level of detail in the assessment of dental carries. For subjects with mixed dentition, a “DMFT” index value was calculated, assessing only the permanent teeth and a “dmft” index value was calculated, assessing only the deciduous teeth. The two values were not added together; instead, the mean value was used for statistical analysis.

The results showed that children involved in competitive sports had lower (better) values for these indices compared to the control group, indicating better oral hygiene and a lower prevalence of periodontal disease and dental caries [50,51]. This is in contrast with previous studies focusing on adult competitive athletes’ oral health [52,53,54,55].

It is important to note that this study had some limitations, including a relatively small sample size and the exclusion of children with certain medical conditions or the recent use of specific medications. Additionally, the study focused on hockey and football players, and the results may not be generalizable to other sports or populations. Moreover, as a cross-sectional study, the temporal link between the outcome and the exposure to sports activity cannot be accurately determined because both are examined at the same time. The differences among groups could be attributed to variations among the study subjects (different dietary or oral hygiene habits). Furthermore, disparities in family social status, household income, dental awareness, and the frequency of dental check-up appointments might also contribute to explaining the results. The relatively limited number of participants, the local recruitment criteria, and the choice of competitive activities (team sports) could be potential sources of bias. Further research is needed to explore the relationship between competitive sports, oral health, and salivary oxidative stress in a larger and more diverse sample. Analyzing the “dose–response effect”, investigating a potential correlation between the number of weekly practice sessions and their effect on salivary oxidative stress could also provide valuable insight into this area.

## 5. Conclusions

This study aimed to investigate the relationship between competitive sports, oral health, and salivary oxidative stress in children. The findings of this study, in line with previous research, suggest that regular physical activity in the form of competitive sports has a positive influence on children’s salivary oxidative stress. The results showed significantly higher levels of glutathione peroxidase (GPX) and total antioxidant capacity (TAC) in children involved in competitive sports compared to the control group, indicating their greater antioxidant capacity and lower oxidative stress. These findings support the notion that regular physical activity can contribute to improved salivary oxidative stress levels in children.

Furthermore, the study also assessed oral health indices, including the Oral Hygiene Index-Simplified (OHI-S), PMA gingival index, and DMF-T index. The results demonstrated that children involved in competitive sports had lower values for these indices, indicating better oral hygiene and a lower prevalence of periodontal disease and dental caries. This contrasts with previous studies focusing on the oral health of adult competitive athletes, which have often shown a higher prevalence of oral health issues. These findings suggest that the positive influence of competitive sports on oral health may be more pronounced in children.

However, it is important to acknowledge the limitations of this study, including the relatively small sample size and the exclusion of children with certain medical conditions or the recent use of specific medications. Additionally, the study focused on hockey and football players, which may limit the generalizability of the findings to other sports or populations. The cross-sectional design of the study also prevents us from establishing a causal relationship between competitive sports, oral health, and salivary oxidative stress. Future research with larger and more diverse samples, as well as longitudinal designs, is needed to further explore the relationship between competitive sports, oral health, and salivary oxidative stress. Additionally, investigating the potential dose–response effect and considering other factors such as dietary habits and socioeconomic status could provide valuable insights into this area.

Taking everything into account, this study provides valuable insights into the relationship between competitive sports, oral health, and salivary oxidative stress in children. The findings suggest that regular physical activity in the form of competitive sports might have a positive impact on children’s oral health by reducing oxidative stress and improving the antioxidant defense mechanisms in the oral cavity. These findings have implications for promoting oral health in children involved in competitive sports and highlight the importance of further research in this area.

## Figures and Tables

**Table 1 healthcare-11-02927-t001:** Demographic data.

Baseline Characteristic	Hockey Group		Football Group		Control Group	
	*n*	%	*n*	%	*n*	%
Gender						
Male	24	41.37	20	36.36	31	51.67
Female	34	58.63	35	63.64	29	48.33
Age						
6 to 12	29	50	27	49.09	30	50
13 to 17	29	50	28	50.91	30	50
Number of training sessions per week						
2	6	10.34	5	9.1	-	-
3–4	23	39.35	28	50.91	-	-
5 or more	29	50.01	22	39.99	-	-

**Table 2 healthcare-11-02927-t002:** Oral health indices’ values.

	Groups	Mean			Median			Standard Deviation (SD)		
Index		H	F	C	H	F	C	H	F	C
OHI-S		0.979	1.472	2.435	1.000	1.83	2.66	0.741	0.501	1.102
PMA		29.683	37.630	43.263	28.101	38.21	45.330	10.059	10.388	11.257
DMF-T		1.328	2.109	3.41	1.012	2.102	3.285	1.033	1.449	1.367

H—all children playing hockey. F—all children playing football. C—all children in control group.

**Table 3 healthcare-11-02927-t003:** Oral health indices’ statistical analysis.

	Groups	OHI-S			PMA			DMF-T		
Statistical Vales		H/C	F/C	H/F	H/C	F/C	H/F	H/C	F/C	H/F
Mean difference		1.4359	0.9573	0.4751	13.4461	5.587	8.0709	1.9483	1.1273	0.7636
SD of differences		1.2303	1.1797	0.9809	14.9185	12.4507	15.6376	2.2589	2.517	1.6326
Minimum sample size		6	11	29	8	34	26	10	34	31
Actual sample size		58	55	55	58	55	55	58	55	55
Two-tailed probability *p*		<0.0001	<0.0001	0.0070	<0.0001	0.0001	0.0030	<0.0001	0.0001	0.012

H—all children playing hockey. F—all children playing football. C—all children in control group.

**Table 4 healthcare-11-02927-t004:** Statistical analysis for salivary oxidative stress biomarkers.

	Comparison Groups	GPX									SOD									TAC								
Statistical Values		H/C	F/C	H/F	H1/C1	F1/C1	H1/F1	H2/C2	F2/C2	H2/F2	H/C	F/C	H/F	H1/C1	F1/C1	H1/F1	H2/C2	F2/C2	H2/F2	H/C	F/C	H/F	H1/C1	F1/C1	H1/F1	H2/C2	F2/C2	H2/F2
Mean difference		−317.9	−255.185	−62.178	−311.174	−251.709	−52.689	−324.745	−253.483	−69.948	32.197	25.926	7.1774	34.66	24.869	8.345	30.187	25.795	4.891	0.635	0.433	0.209	0.62	0.415	0.206	0.684	0.465	0.205
SD of differences		31.109	54.954	84.329	58.618	52.542	74.786	51.601	60.705	76.417	16.019	20.848	22.146	17.305	22.881	23.548	14.941	20.175	16.74	0.548	0.561	0.293	0.588	0.629	0.337	0.478	0.464	0.313
Minimum sample size		4	11	48	10	11	24	4	5	15	10	19	64	10	19	54	8	11	78	22	24	14	16	22	19	10	8	17
Actual sample size		58	55	55	29	27	27	29	28	28	58	55	55	29	27	27	29	28	28	58	55	55	29	27	27	29	28	28
Two-tailed probability *p*		<0.0001	<0.0001	<0.0001	<0.0001	<0.0001	0.0011	<0.0001	<0.0001	<0.0001	<0.0001	<0.0001	0.0197	<0.0001	<0.0001	0.077	<0.0001	<0.0001	0.1338	<0.0001	<0.0001	<0.0001	<0.0001	0.0021	0.0038	<0.0001	<0.0001	0.0018

H—all children playing hockey; H1—all children playing hockey aged 6–12; H2—all children playing hockey aged 13–17. F—all children playing football; F1—all children playing football aged 6–12; F2—all children playing football aged 13–17. C—all children in control group; C1—all children in control group aged 6–12; C2—all children in control group aged 13–17.

## Data Availability

The data presented in this study are available upon request from the corresponding author. The data are not publicly available due to privacy reasons.

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
