# Peer review of "The Impact of Competitive Sports on Oral Health: Exploring Their Relationship with Salivary Oxidative Stress in Children"

_healthcare, 2023, doi:10.3390/healthcare11222927_

Round 1
Reviewer 1 Report
Comments and Suggestions for Authors
Dear Authors
I appreciate the topic of the paper. It is a very interesting point of view.
I suggest to modify some parts of the article in order to make it easy to read for the readers.
Introduction: Please be more concise. Reduce to 3/4 paragraphs. Paragraphs describing each salivary marker are too long, please merge them.
Introduction: in the last paragraph do not anticipate conclusions (lines 154-159).
Materials and Methods: Indicate the clinicians that performed the clinical examinations and the saliva collection (line 189-192)
Please remove lines 184-188.
Lines 205-214 should be removed from ' Materials and Methods'. I suggest to insert these informations in Discussion.
Results: Results should be reported in a well organized table. Please delete Paragraphs 3.2. Avoid reporting data in text. Insert a table (or even 2) that summarizes all the results and the statistical analyses. Legend of the table should be reported under each table.
Best regards
Reviewer 2 Report
Comments and Suggestions for Authors
The manuscript is of excellent quality and addresses a topic relevant to oral health conditions in a specific population - children and adolescents who play sports competitively. In this sense, some questions that potentially did not alter the analysis: did the authors consider removing those who participated in less than 2 sessions per week, considering the object of study? Has this adjustment been performed?
Could the results be changed depending on frequency? (dose response effect?).
I reinforce the excellent quality of the work and congratulate the team.
Comments on the Quality of English LanguageVery good
Reviewer 3 Report
Comments and Suggestions for Authors
The introduction is too extensive and does not include rationale for the study. I recommend providing more information on how sports directly or indirectly affect oxidative stress levels of children. The last paragraph of the introduction should belong in methods section. I would like to know why the subjects were categorized to football and hockey groups. Do the sports vary in their relationship to oxidative stress? Line 179-180 : please explain how did you determine 7 subjects do not alter the findings? For the cross sectional studies, multiple regression analyses should be used to provide inferential analysis.
Comments on the Quality of English LanguageOverall, the english is good; however, I noticed some typos.
Reviewer 4 Report
Comments and Suggestions for Authors
This study holds relevance in grasping the importance of advocating for children's oral health in conjunction with their physical activity levels. It underscores the value of comprehending salivary oxidative stress in children through the non-invasive method of saliva sampling.
1. Although recruitment process of the subjects was not described in detail for both competitive sports and control groups.
2. Would the lower DMFTs in competitive sports children, be due to more regular dental checks/visits by families or sports club? What about social status of families, better dental awareness, toothbrushing habits etc?
Round 2
Reviewer 3 Report
Comments and Suggestions for Authors
I appreciate making changes according to my feedback